# Potentially inappropriate medication uses and associated factors among elderly primary health care clinics attendees: A call to action

Esra'a Samara[1], Zaher Nazzal[2]*, Shayma Naghnaghia[1], Rowa' AL-Ramahi[3]*

**1** Faculty of Medicine and Health Sciences, Department of Medicine, Family medicine resident, An-Najah National University, Nablus, Palestine, **2** Faculty of Medicine and Health Sciences, Department of Medicine, Consultant Community Medicine, An-Najah National University, Nablus, Palestine, **3** Faculty of Medicine & Health Sciences, Department of Pharmacy, Professor of Clinical Pharmacy, An-Najah National University, Nablus, Palestine

☺ These authors contributed equally to this work.
* znazzal@najah.edu (ZN); rawa_ramahi@najah.edu (RA-R)

## Abstract

### Background

Polypharmacy is a significant risk factor for using potentially inappropriate medication (PIM), which is using drugs with more risks than benefits, especially for elders. This study aimed to estimate the prevalence of PIM using Beers Criteria, polypharmacy, and their related risk factors.

### Methods

A descriptive cross-sectional study was conducted in West Bank primary health care clinics (PHC)from December 2021 to March 2022. Data were collected from PHC clinic attendees aged 65 and above via an interviewer-administered questionnaire and a review of their medical records. We used the Beers Criteria 2019 update to identify PIM and performed a multivariable analysis to determine its associated factors.

### Results

The study included 421 older people (197 men and 224 women) with an average age of 73.6 years. The prevalence of polypharmacy was 75.1% (95%CI: 70.6%-79.1%), with an average of six medications dispensed per patient. On the other hand, PIM was identified among 36.8% of the study participants (CI:95%CI: 32.2%- 41.6%). Sulfonylureas were the most common (24.2%) reported PIM, followed by peripheral alpha-blockers (4.3%), non-steroidal anti-inflammatory drugs (3.1%), proton pump inhibitors (2.9%), and central nervous system medications (2.1%). Hyperpolypharmacy (> 10 drugs) [aOR = 4.1, 95%CI: 1.6–10.7], polypharmacy [aOR = 2.8, 95%CI: 1.4–5.4], and Diabetes [aOR = 3.5, 95%CI: 2.0–6.0] are the main associated factors of PIM.

### Conclusion

This study found that over one-third of the older people attending PHC clinics have PIM, with polypharmacy and Diabetes being the main predicting variables. Improving physicians'

**Data Availability Statement:** All relevant data are within the paper and its Supporting Information files.

**Funding:** The authors received no specific funding for this work.

**Competing interests:** The authors have declared that no competing interests exist.

awareness of clear and specific PIM lists can reduce the number of PIM prescribed and decrease their impact.

## Introduction

People around the world are living longer than ever before. The percentage of older people is expected to double by 2050 [1]. The term elderly has been used to describe people above 65 years old. The presence of concomitant diseases and complex prescriptions in this age group gave rise to polypharmacy, defined as taking five or more medications while taking ten or more is considered excessive [2]. Multiple morbidities, older age, lower education level, and frequent outpatient clinic visits are well-known predictors of polypharmacy [2].

Besides polypharmacy, the change in medication's pharmacokinetics and pharmacodynamics due to aging's physiologic changes, such as reduced renal function, high-fat mass, and low muscle mass, all played a significant role in increasing drug toxicity and adverse effect in the elderly contributed to the emergence of the so-called potentially inappropriate medications (PIM) [3]. Medication is considered PIM when its risk is more significant than its benefit, prescribed in an inappropriate dose, or for an unsuitable duration [4]. PIM among the elderly is currently a global concern since its negative impact on quality and cost on health, as it leads to unnecessary wastage of health resources by increasing the risk of hospitalization, new additional medication prescriptions, and increasing morbidity and mortality rate [4].

Several tools have been used to screen PIM in the elderly to improve medication selection and reduce their adverse effect; the most commonly used tools are the European STOPP/START criteria and the American Beers criteria [5]. Beers criteria are widely employed in people above 65 and can be used in all healthcare settings except palliative and hospice care. It was developed in 1991 by Mark Beers and has been updated every few years since 2011; the most recent update was in 2019, published by the American Geriatric Society (AGS) [6].

The criteria are organized into ten tables; the first indicating evidence quality and strength and the second to six indicating the following: PIM in the elderly in general, drug-disease, drug-drug interactions, those to be used with caution in the elderly, and those to avoid or modify based on renal function, respectively, the seventh list medications with potent anticholinergic properties used in the previous tables. The last three tables summarize the changes made to 2015 criteria as follows: the eighth and ninth indicate drugs that have been withdrawn or added since the 2015 criteria, as well as those that are no longer on the market, while the tenth includes medications whose recommendations have been updated [6]. It is noteworthy that this tool, similar to the other STOPP/START tool, cannot be used as a definitive tool for making a medical decision, instead acts as a decision supporter along with clinical judgment, common sense, and weighting the benefit-risk in each patient, to determine the appropriate prescribing [6]. Besides, Beers criteria reflect the United States drug market, while STOPP/START criteria reflect the European market, which may not reflect other countries' drugs market [7].

The prevalence of PIM and the most frequently prescribed PIMs vary depending on the population, study setting, type, and version of evaluation criteria. Reviewing previous studies in a primary care settings, according to Beers Criteria using various versions, the most common PIM were proton pump inhibitors (PPIs) [8–11], long acting sulfonylureas [7,11], nonsteroidal anti-inflammatory drugs (NSAIDs) [8–10] benzodiazepines [8,12] and central nervous systems drugs [12].

In Palestine, the elderly account for around 3.3% (1.5% male and 1.8% female) of the population and are rising [13]. They have a high prevalence of chronic diseases, including but not

limited to diabetes, hypertension, congestive heart failure, atrial fibrillation, and coronary heart disease. These conditions are commonly managed with medications such as furosemide, hydrochlorothiazide, atorvastatin, levothyroxine, lisinopril, metoprolol, simvastatin, atenolol, amlodipine, and metformin [14]. The primary objective of this study is to measure the prevalence of PIM use among Palestinian elderly patients attending PHC clinics and to study the associated factors using the latest version of the AGS 2019 Beers Criteria.

## Methodology

### Study design

An observational descriptive cross-sectional study was conducted among elderly people attending primary health care (PHC) clinics of the Ministry of Health (MOH). PHC meets most individuals' lifelong health needs and provides various healthcare services, including health promotion, disease prevention, treatment, rehabilitation, and palliative care. In Palestine, the MOH manages up to two-thirds of PHC clinics, while the remainder is managed by non-governmental organizations [14]. In addition, most of the older people in Palestinian (two-thirds) had governmental insurance rather than private or no insurance [15]. We gathered data from patients through interviews and reviewed their medical files between December 2021 to March 2022 in the three main cities of the northern West Bank of Palestine; Nablus, Jenin, and Tukaram. We selected six PHC centers in each district, making a total of 18 clinics involved in the study. All records collected were assessed for PIM using the Beers Criteria 2019 update.

The study included all patients aged 65 and above who had been taking medications for more than two weeks and had visited PHC clinics during the study period, regardless of disease. A minimum sample size of 387 was calculated using the OpenEpi software, and the equation n = [DEFF*Np(1-p)]/ [(d2/Z21-/2*(N-1) +p*(1-p)], with an expected outcome of roughly 50%, precision <0.05, a 95% confidence interval, and a DEFF = 1.

### Data collection and measures

We used an interviewer-administered questionnaire for data collection. It consists of three sections; the first is about demographic data such as age, gender, weight, residency, marital status, monthly income, educational level, and Living arrangement. The second section was more about the patient's medical history, including a list of medications used during the last two weeks by their main ingredient and dosage form, whether these were prescriptions or over-the-counter medications, and the places of prescriptions. Finally, the third section was about the history of hospitalization in the previous two years and their average PHC clinic visits, which should be one every three months according to PHC clinics' chronic disease protocols in Palestinian's MOH. While most of the data were available in patients' files, interviews helped gather more information about medications from other places and to double-check patients' medications. The researchers created the questionnaire after reviewing related literature [16–18]. After three experts in the field reviewed it, we conducted a pilot study on 40 subjects from the study population. The interviews were conducted by two of the authors (IS and SN), who are family physicians and have been trained to conduct interviews.

Then, we reviewed the patient's medical records to double-check diseases, medications, and other information, such as their weight and creatinine level within the previous three months. We calculated creatinine clearance (CrCl) to evaluate medications based on the Beers Criteria, using the Crockcroft Gault formula: (140-age)*(weight (kg))*(0.85 if female)/(72*cr, mg/dl). Polypharmacy was evaluated based on the number of medications used by each patient. It is defined as the concurrent use of five or more medications over two weeks [19]. To identify

patients with PIM, we reviewed medications for each participant using the 2019 updated AGS Beers Criteria [20]. Any medication found to have moderate to strong recommendations to be avoided in the elderly was considered PIM. Medicines containing two compounds were regarded as two separate medications, and each constituent was to be investigated separately to determine the PIM for each. The prevalence of PIM was calculated as the number of participants who used at least one PIM divided by the total number of participants.

## Data analysis

Questionnaires were checked for completeness before being entered into Microsoft Office Excel and coded and analyzed using the IBM SPSS Statistics for Windows, version 21 (IBM Corp., Armonk, NY, USA). This study had no missing data due to an interviewer-administered questionnaire and a check of participants' medical records. Descriptive statistics such as frequency and percentages and their 95% confidence intervals are used to describe patient characteristics, polypharmacy prevalence, and PIM. We used the chi-square test to examine the relationship between independent variables (age, gender, residency, number of prescription locations, working status, income, and educational level) and polypharmacy and PIM. The logistic regression model was employed to analyze variables independently associated with PIM by including all variables found to be significant on bivariable analysis and those considered important in the literature. P values <0.05 were considered statistically significant.

## Ethical consideration

The study was approved by the Institutional Review Board of An-Najah National University [Reference #: Farm. Med. Dec. 2021/2]. The Palestinian Ministry of Health approved the study and granted access to the data and permission to interview the patients. Attendees at the elderly PHC clinics were invited to participate in the study voluntarily, and those who agreed signed informed written consent. Participants were interviewed in a private room, knowing their information would be used for research purposes only and that confidentiality would be guaranteed. To ensure the anonymity of each participant, a random number is assigned to their questionnaire and medical records.

## Results

### Participants' descriptive characteristics

A total of 421 geriatric patients of PHC clinic attendants participated in this study. Their ages ranged from 65 to 100 years, with an average age of 73 ±6.2. Almost half of them (46.8%) were male, 66.3% were married, and 59.1% had under high school education level. Most of them (90.5%) were not working, and most (89.1%) lived with their families. Almost 60% of the participants have admitted that they get their medication from more than one place, and the majority (87.2%) visited PHC clinics more than once in the last three months (Table 1). Regarding medical history, 98.3% of the study's participants have chronic diseases. The most common disease was hypertension (84.8%), followed by diabetes (56.3%) and heart diseases (36.6%); one-fourth of them has more than three chronic diseases at once (Table 1).

### Medications

The average number of total medications used per patient was six, ranging from one to 15. About two-thirds of them (66.7%) use 5–9 medications. Aspirin was the most frequently used medication among the study participants (84.6%), followed by atorvastatin (73.9%), renin-angiotensin system Inhibitors (53.7%), diuretics (48%), and metformin (43.5%) (Table 2). The

**Table 1. Demographic characteristics and medical history of study participants (n = 421).**

| Characteristics | Frequency | Percentages |
|---|---|---|
| **Gender** | | |
| Male | 197 | 46.8% |
| Female | 224 | 53.2% |
| **Age groups** | | |
| 65-69years | 144 | 34.2% |
| 70–74 years | 135 | 32.1% |
| 75–79 years | 71 | 16.9% |
| ≥80 years | 71 | 16.9% |
| **Residency** | | |
| Urban | 228 | 54.2% |
| Rural | 193 | 45.8% |
| **Working status** | | |
| Working | 40 | 9.5% |
| Not working | 381 | 90.5% |
| **Marital status** | | |
| Married | 279 | 66.3% |
| Unmarried | 142 | 33.7% |
| **Education level** | | |
| Under high school | 249 | 59.1% |
| High school | 75 | 17.8% |
| University or diploma | 97 | 23.0% |
| **Income per month** | | |
| Less than 600$ | 255 | 60.6% |
| 600–1199 $ | 144 | 34.2% |
| ≥1200$ | 22 | 5.2% |
| **Living arrangement** | | |
| Alone | 46 | 10.9% |
| With family | 375 | 89.1% |
| **Place of prescription** | | |
| One source | 168 | 39.9% |
| More than one source | 253 | 60.1% |
| **Average** PHC clinics visits **for the last three months** | | |
| ≤ one visit | 54 | 12.8% |
| > one visit | 367 | 87.2% |
| **Chronic Disease** *(Yes)* | 414 | 98.3% |
| Hypertension | 357 | 84.8% |
| Diabetes | 237 | 56.3% |
| Heart diseases | 154 | 36.6% |
| Stroke | 40 | 9.5% |
| **Number of chronic diseases** | | |
| One or less | 65 | 15.4% |
| Two | 125 | 45.1% |
| Three | 124 | 29.5% |
| More than three | 107 | 25.4% |
| **Hospitalization during the past two years** | | |
| Yes | 108 | 25.7% |
| No | 313 | 74.3% |

**Table 2. Distribution of the most commonly prescribed medications among elderly PHC clinics patients and reported side effects.**

| Medication | Frequency | Percentages |
|---|---|---|
| **Anti-platelet** | | |
| Aspirin | 356 | 84.6% |
| **Lipid-lowering-agent** | | |
| Atorvastatin40mg | 311 | 73.9% |
| **Hypoglycemic agent** | | |
| Metformin | 183 | 43.5% |
| **Antihypertensive drugs** | | |
| Renin-angiotensin system inhibitors* | 226 | 53.7% |
| **Proton Pump Inhibitors (PPIs)** | 99 | 23.5% |
| **Number of medications** | | |
| Less than five | 105 | 24.9% |
| Five to nine | 281 | 66.7% |
| Ten or more | 35 | 08.3% |

\* Patients who used many subclass drugs were regarded as taking one main-class drug; hence there is a discrepancy between subclasses and main-class pharmaceuticals.

results showed polypharmacy was reported among 75.1% of the participants (95%CI: 70.6%-79.1%).

## Potentially inappropriate medication and related factors

The prevalence of PIM was 36.8% (95%CI:32.2%- 41.6%), the majority having one potentially inappropriate medication. Long-acting sulfonylurea, including glimepiride and glyburide, was the most prescribed PIM among the study's participants (24.2%), followed by peripheral alpha-blockers (4.3%), NSAIDs (3.1%), and PPIs (2.9%) (Table 3).

On bivariable analysis, PIM was found to be significantly more likely among elderly PHC clinics patients who received medications from multiple sources, had a higher number of chronic diseases, especially Diabetes, and had polypharmacy. Multivariate analysis was used to identify factors independently associated with PIM. Polypharmacy was found to be the strongest predicting factor, with elderly patients who took more than ten medications being four times more likely to have PIM [aP-value = .0054, aOR = 4.1, 95%CI: 1.6–10.7], while those who took five to nine medications were 2.8 more likely [aP-value = .003, aOR = 2.8, 95%CI: 1.4–5.4]. Those with DM were 3.5 times more likely to have PIM compared to the elderly without DM; [aP-value = < .001, aOR = 3.5, 95%CI: 2.0–6.0]. On the other hand, older adults with hypertension are less likely to have PIM; [aP-value = .007, aOR = .39, 95%CI: .19 - .76] (Table 4).

## Discussion

The primary objective of this study was to assess the prevalence of PIM in primary care using the most recent revision of the Beers Criteria (the 2019 AGS criteria). More women (54.5%) than men were found in our study, consistent with the census of the elderly Palestinian population [14]. In addition, HTN and Diabetes were found to be the most common non-communicable diseases among study participants, as in previous local studies [21,22]. Over one-third of elderly patients attending PHC clinics in our study were found to use at least one PIM. This falls within the scope of PIM prevalence worldwide, where PIM prevalence using the 2019

**Table 3. Classes of the most commonly used potentially inappropriate medications among elderly PHC patients.**

| Medication class and name | FREQUENCIES | | PERCENTAGE % |
|---|---|---|---|
| **Long-acting sulfonylurea** | 102 | | 24.2% |
| Glimepiride | | 97 | |
| Glyburide | | 5 | |
| **Proton pump inhibitors > eight weeks** | 12 | | 2.9% |
| Omeprazole | | 11 | |
| Esomeprazole | | 1 | |
| Lansoprazole | | 0 | |
| pantoprazole | | 0 | |
| **Alpha-blockers** | 24 | | 4.3% |
| Doxazosin | | 22 | |
| Tamsulosin | | 2 | |
| **NSAID** | 13 | | 3.1% |
| **NERVOUS SYSTEM MEDICATIONS** | 9 | | 2.1% |
| anti-parkinson * | | 6 | |
| BENZODIAZEPineS | | 1 | |
| others | | 2 | |
| **FIRST-GENERATION ANTI-HISTAMINE** | 6 | | 1.4% |
| **DESMOPRESSIN** | 1 | | 0.2% |
| **AMIODARONE** | 1 | | 0.2% |
| **DIGOXIN** | 7 | | 1.9% |

*Include benztropine and trihexyphenidyl.

Beers Criteria was 34.4% in the USA [23], 49.2% in Jordan [24], 62.2% in both India and Qatar [16,17], and 68.8% in Spain [17].

Long-acting sulfonylurea was found to be the most frequent PIM in our study (24.2%). Taking into consideration the fact that 56.3% of study participants have Diabetes and half of the diabetic patients in our study are on sulfonylurea, which is still one of the available first-line hypoglycemic agents that health insurance can cover in this country, even though its significant side effect is hypoglycemia, which can last for many hours and may necessitate hospitalization [25]. Globally, recommendations are against considering sulfonylurea as one of the first-line treatments of Diabetes, especially in the elderly, who may live alone or have a limited social support system and whose food intake is unreliable [25]. This emphasizes the need to inform healthcare providers and policymakers about alternative lower-risk hypoglycemic options to sulfonylurea that may be covered by health insurance for people in that age group.

The second most frequent PIM was peripheral alpha-blockers (4.3%), which are being used to treat various conditions, most commonly benign prostatic hyperplasia and high blood pressure, but according to Beers Criteria, healthcare providers should avoid using them for hypertension treatment in the elderly because of its vasodilatory effect, as one of its side effects is orthostatic hypotension, which can lead to bone fracture, especially in the elderly [26]. Instead, they should choose other, more safe alternatives [26]. NSAID was the third most frequent PIM used by the study participants, and it is used widely to treat multiple inflammatory and pain conditions which are commonly seen in this age group; while it's effective in treating such conditions, its most common side effects could be gastrointestinal bleeding which may be fatal in elderly and may need urgent care, so health care provider has to consider this risk before choosing what drug to use for pain management in elderly.

Compared with previous studies conducted in PHC settings utilizing various versions of the Beers criteria, it was shown that long-acting sulfonylureas [7,11] and NSAIDs [8–10]

**Table 4. Bivariable and multivariable analysis of variables related to potentially inappropriate medication use.**

| | Potentially inappropriate medications | | | Multivariable analysis | |
| --- | --- | --- | --- | --- | --- |
| | **Yes** | **No** | **p-value*** | | |
| | **Frequency (%)** | **Frequency (%)** | | **aP-value** | **aOR (95%CI)** |
| **Gender** | | | 0.207 | | |
| Male† | 68(34.5%) | 129(65.5%) | | .610 | 1.1 (.70–1.7) |
| Female | 87(38.8%) | 137(61.2%) | | 1 | |
| **Source of Medication** | | | .026 | | |
| From one source† | 52(31.0%) | 116(69%) | | 1 | |
| From more than one source | 103(40.7%) | 150(59.3%) | | .368 | 1.2 (.77–2.0) |
| **Age groups** | | | .975 | | |
| 65-69years | 55(38.2%) | 89(61.8%) | | 1 | |
| 70–74 years | 49(36.3%) | 86(63.7%) | | .741 | 1.1 (.64–1.9) |
| 75–79 years | 26(36.6%) | 45(63.4%) | | .661 | 1.2 (.67–2.0) |
| ≥80 years | 25(35.2%) | 46(64.8%) | | .834 | 1.1 (.61–2.0) |
| **Residency** | | | 0.310 | | |
| Urban | 81(35.5%) | 147(64.5%) | | – | — |
| Rural† | 74(38.3%) | 119(61.7%) | | | |
| **Working status** | | | 0.169 | | |
| Working | 18(45.0%) | 22(55.0%) | | – | — |
| Not working | 137(36.0%) | 244(64.0%) | | | |
| **Marital status** | | | 0.132 | | |
| Married | 97(34.8%) | 182(65.2%) | | – | — |
| Unmarried | 58(40.8%) | 84(59.2%) | | | |
| **Education level** | | | 0.291 | | |
| Under high school† | 99(39.8%) | 150(60.2%) | | 1 | |
| High school | 23(30.7%) | 52(69.3%) | | .153 | .64 (.34–1.2) |
| University or diploma | 33(34.0%) | 64(66.0%) | | .809 | .93 (.52–1.7) |
| **Income per month** | | | 0.876 | | |
| Less than 600$ | 94(36.9%) | 161(63.1%) | | – | — |
| 600–1199$ | 54(37.5%) | 90(62.5%) | | | |
| ≥600 | 7(31.8%) | 15(68.2%) | | | |
| **Living arrangement** | | | 0.132 | | |
| Alone | 13(28.3%) | 33(71.7%) | | – | — |
| With family | 142(37.9%) | 233(62.1%) | | | |
| **Average PHC clinics visits for the last three months** | | | 0.091 | | |
| ≤ one visit† | 15(27.8%) | 39(72.2%)) | | .545 | 1.3 (.60–2.7) |
| > One visit | 140(38.1%) | 227(61.9%) | | | |
| **Number of chronic diseases/people** | | | 0.003 | | |
| One or less† | 16(24.6%) | 49(75.4%) | | 1 | |
| Two | 41(32.8%) | 84(67.2%) | | .597 | .80 (.35–1.9) |
| Three | 44(35.5%) | 80(64.5%) | | .482 | .24 (.29–1.8) |
| More than three | 54(50.5%) | 53(49.5%) | | .947 | 1.1 (.39–2.7) |
| **Hypertension** | | | 0.024 | | |
| Yes | 123(34.5%) | 234(65.5%) | | .007 | .37 (.19-.76 |
| No† | 32(50.0%) | 32(50.0%) | | 1 | |
| **Diabetes Mellitus** | | | < .001 | | |
| Yes | 121(51.1%) | 116(48.9%) | | < .001 | 3.5 (2.0–6.0) |
| No† | 34(18.5%) | 150(81.5%) | | 1 | |

*(Continued)*

**Table 4.** (Continued)

| | Potentially inappropriate medications | | p-value* | Multivariable analysis | |
|---|---|---|---|---|---|
| | **Yes** | **No** | | aP-value | aOR (95%CI) |
| | **Frequency (%)** | **Frequency (%)** | | | |
| **Heart diseases** | | | 0.323 | | |
| Yes | 54(35.1%) | 100(64.9%) | | – | — |
| No | 101(37.8%) | 166(62.2%) | | | |
| **Number of medications** | | | < .001 | | |
| Less than 5† | 17(16.2%) | 88(83.9%) | | | |
| | 119(42.3%) | 162(57.7%) | | .003 | 2.8 (1.4–5.4) |
| More than 10 | 19(54.3%) | 16(45.7%) | | .005 | 4.1 (1.6–10.7) |

†*Reference group*

*\*Chi-squared test,**a**P-value: adjusted P-value, **aOR**: adjusted Odds Ratio, **CI**: Confidence Interval.*

ranked among the three most frequently seen PIMs in Jordan, Saudi Arabia, and Brazil. On the other hand, CNS drugs [12] and benzodiazepines [8,12] were identified as commonly prescribed PIMs. However, it is essential to note that our study did not investigate these specific drug categories, limiting our ability to establish a direct correlation. Multiple studies have consistently identified PPIs as one of the most commonly utilized PIMs in older populations [8–11].

PPIs are prescribed in the elderly or used as an over-the-counter medicine to protect, treat, or maintain a variety of common gastrointestinal diseases in the elderly [27]. They are considered potentially inappropriate medication if they are used for more than eight weeks, except for high-risk patients [27]. In our study, while a quarter of the patients were on PPIs, most of them (85% of the study's participants) were taking low-dose aspirin simultaneously, which increased the risk of gastrointestinal bleeding. In this case, PPIs are not considered as PIM [28]; however, this raises questions about whether aspirin is misused among these patients, highlighting the need for further investigation and follow-up by competent authorities.

Differences in the study's setting and evaluation tools may yeild different results regarding PIM's prevalence and the most frequent PIM medications. A local study conducted among hospitalized Palestinian patients to check for potentially inappropriate prescriptions using STOPP/START criteria reported a more significant proportion of potentially inappropriate prescriptions (PIM) than we did (54.6%), with drug therapy duplication being the most prevalent PIM compared to Long-acting sulfonylurea in our study [29]. This comparison demonstrates the need to conduct future studies using both tools to evaluate medications for the elderly and perhaps to develop a national tool that suits the market and is adopted as a reference by practitioners [7].

Consistent with previous studies [11,16,30,31], multivariate logistic regression has shown that the number of medications (polypharmacy) and the presence of Diabetes are significantly associated with PIM prescription. The finding that Diabetes is a significantly associated factor of PIM prescription may be explained by the fact that sulfonylureas are one of the main potentially inappropriate medicines in the Beers Criteria, and they are still used to treat Diabetes in many cases. In contrast, to many previous studies and unexpectedly, age, gender, and the number of chronic diseases were found to have no significance in predicting PIM, and knowing that 84.8% of study participants who have had hypertension are found to have less risk of PIM compared to those who do not, even though peripheral alpha-blockers, which were used among hypertensive patients, were the second PIM found in our study. The differences in the

finding of this study compared to previous studies could be attributed to the difference in sample size, population, and sitting of these studies. It is worth mentioning that other studies used electronic records of thousands of participants, some were retrospective studies studying people over the years, and most were inpatient studies.

In this study, polypharmacy was a significantly associated factor of PIM. Many drugs may result in an improper combination of medications or a decline in renal function, which can turn some medications from appropriate to inappropriate and increase the risk of PIM. Almost three-quarters of the study participants were found to have polypharmacy, which is considered very high when compared to the global polypharmacy prevalence in PHC clinics, which ranges from 5% to 77% [31], and higher than previous local studies of the same population (41%) [32]. The well-known determinants of polypharmacy mentioned in previous studies were old age, low level of health education, poor health conditions, and multiple outpatient department visits [2]. While in our study, polypharmacy was found to be higher among participants with more frequent visits to PHC clinics, who get medication from more than one place, with low income, and who have a history of multiple co-morbidities, mostly DM, compared to a previous local study in primary care females aged 60–69, with multiple diseases with some regional difference in drug utilization, found to have more prevalent polypharmacy [22]. Differences in sample size, sitting, and time of studies may refer to this difference in polypharmacy-associated factors. Higher health care costs, medication nonadherence, and decreased elderly's functional status are well-known outcomes of polypharmacy other than PIM [33].

Another notable finding is that most participants visited PHC clinics more frequently than planned, and more than half admitted getting their medication from multiple sources. This frequent PHC clinic visits factor largely contributed to the prevalence of polypharmacy, and policymakers should evaluate and address them to improve healthcare coordination and minimize costs.

To our knowledge, this is the first study in Palestine to report the PIM prescription using Beers Criteria 2019. Besides, it is the first report that compares polypharmacy and PIM in primary health care, knowing that most health care practitioners in Palestine have low to moderate awareness of PIM and Beers Criteria [34]. However, some limitations should be taken into consideration. Although the numerical count-based definition of polypharmacy utilized in our study is widely accepted, the fact that it is not widely used in practice may be one of the study's limitations. On the other hand, we could not assess our participants' psychiatric medications since many of them were classified as PIM under Beers Criteria. Furthermore, psychiatric medications are dispensed in separate files from general medicine clinics, making it difficult to determine whether study participants received any of them; noticing that our focus in this study was more on medication to be avoided in the elderly and not looking for those to be used with caution in the elderly (Table 4 in Beer's criteria), considering that two points prevalence of PIM in Palestine could be higher than what was found in this study.

## Conclusion

Our study showed that polypharmacy is high in older adult patients attending Palestine's PHC clinics, and many PIM that should be avoided were prescribed to those elderly. The most common PIM was long-acting sulfonylurea. Diabetes and the number of medications (polypharmacy) were significantly associated with PIM. We recommend that policymakers adopt interventions that educate and train healthcare providers at all levels on PIM in general and Beers Criteria and its updates, as well as develop a clear plan for resolving multi-sources of medication prescriptions. This can be accomplished by applying the family medicine program to PHC clinics where good follow-up and a doctor-patient relationship may be the key

solution; we insist on conducting studies on Aspirin use in the elderly to find out if there is any misuse and ultimately provide less risky alternatives to the PIM mentioned earlier in this study.

## Supporting information

**S1 Checklist. STROBE statement—checklist of items that should be included in reports of observational studies.**
(DOCX)

**S1 File.**
(XLSX)

## Acknowledgments

We thank the Palestinian Ministry of Health and the head of Primary Health Care directorates in the Northern West Bank for the help and contribution they have given us in facilitating our study. We thank the study participants for their efforts and time contributing to the study.

## Author Contributions

**Conceptualization:** Esra'a Samara, Zaher Nazzal, Shayma Naghnaghia, Rowa' AL-Ramahi.

**Data curation:** Esra'a Samara, Shayma Naghnaghia.

**Formal analysis:** Zaher Nazzal, Rowa' AL-Ramahi.

**Methodology:** Esra'a Samara, Zaher Nazzal, Shayma Naghnaghia, Rowa' AL-Ramahi.

**Project administration:** Zaher Nazzal, Rowa' AL-Ramahi.

**Software:** Zaher Nazzal.

**Supervision:** Zaher Nazzal, Rowa' AL-Ramahi.

**Validation:** Esra'a Samara.

**Visualization:** Esra'a Samara.

**Writing – original draft:** Esra'a Samara, Shayma Naghnaghia.

**Writing – review & editing:** Zaher Nazzal, Rowa' AL-Ramahi.

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
