## [Decision Letter · Decision Letter 0]

24 Jul 2023

PONE-D-23-10547Potentially inappropriate medication uses and associated factors among elderly primary health care clinics attendees: a call to actionPLOS ONE

Dear Dr. Nazzal,

Thank you for submitting your manuscript to PLOS ONE. After careful consideration, we feel that it has merit but does not fully meet PLOS ONE’s publication criteria as it currently stands. Therefore, we invite you to submit a revised version of the manuscript that addresses the points raised during the review process.

We look forward to receiving your revised manuscript.

Kind regards,

Dr. Muhammad Eid Akkawi

Academic Editor

PLOS ONE

Journal Requirements:

"The author(s) received no financial support for this article's research, authorship, and/or publication."

"The authors declare that they have no competing interests in this study."

Reviewers' comments:

Reviewer's Responses to Questions

**Comments to the Author**

1. Is the manuscript technically sound, and do the data support the conclusions?

Reviewer #1: Yes

Reviewer #2: Yes

2. Has the statistical analysis been performed appropriately and rigorously? 

Reviewer #1: I Don't Know

Reviewer #2: Yes

3. Have the authors made all data underlying the findings in their manuscript fully available?

Reviewer #1: Yes

Reviewer #2: No

4. Is the manuscript presented in an intelligible fashion and written in standard English?

Reviewer #1: Yes

Reviewer #2: No

5. Review Comments to the Author

Reviewer #1: COMMENTS FOR MANUSCRIPT NUMBER PONE-D-23-10547

It is an interesting work done on the prevalence polypharmacy and probably inappropriate medications (PIM) and it associated risk factors. I can relate well with it but couldn’t do much to confirm the appropriateness of the statistical analysis employed. Hence, I have a few suggestions to make.

1. There are a few grammatical errors that should be corrected.

2. The spelling of some of the medicines should be corrected. Eg. Benzodiazepine and Amiodarone. It would be appropriate if generic names were used for the specific examples of medicines names used in Table 3.

3. Briefly describe how the Beer’s criteria is used in such analysis instead of or in addition to the changes or update made to the 2015 version in your introduction.

4. Does it have any setbacks?

5. In all, how many PHC clinics did you conduct this study?

6. Did all participants have creatinine level in their medical folders? How recent were they?

7. Nothing was said about the creatinine level and weight data collected in the patient’s medical records in the discussion. Or was it only to aid in the evaluation of medicines for Beer’s criteria?

8. Apart from the Beer’s criteria, did you consider any current reference for medicines classification or status recommendations in terms of PIM?

9. Is any of the authors a practicing Pharmacist?

10. According to line 179, I think it wasn’t surprising that long-acting sulfonylureas was found to be the most frequent PIM.

11. Could you further clarify why psychiatric medications could not be assessed as PIM?.

Reviewer #2: 

General: English language editing is needed.

Methodology:

Is a medication with two ingredients considered as one or two medications?

Results:

“The average number of total medications used per patient was six, ranging from zero to 15”. How were patients with zero medications included in the study? The inclusion criteria say: The study included all patients ≥ 65 years who had been taking medications for more than two weeks.

Table 2: Is the number of patients using hypoglycaemic agents identical to that who using antihypertensive agents? both are 226?

Table 3: Add ‘first-generation’ antihistamines.

Multiple regression: I believe that the multiple regression model should not be limited to variables that showed significant bivariable association. This is because some of these variables can still affect the tendency of having the independent variable (PIM) especially if there is a significant difference in the distribution of this variable between people with and without PIM.

Table 4: Related to the above comment, the distribution of the independent variables (e.g., male/female) should be presented based on the dependent variable. i.e., in PIM group, how many males and females were among these 155 patients? The sum should be 100%.

Discussion:

Compare the results of the most encountered PIMs (SU, alpha blockers and NSAIDs) with other studies.

The results should be compared to other studies done in Palestine even those used criteria other than Beers’ (e.g., https://www.dovepress.com/evaluating-inappropriate-medication-prescribing-among-elderly-patients-peer-reviewed-fulltext-article-CIA).

“We believe that the findings of this study are generalizable to the elderly Palestinians attending PHC centers because of the relatively large sample size drawn from different districts of Palestine”

As the sampling technique was convenient rather than stratified randomized nationwide sampling, the above assumption is not valid.

6. PLOS authors have the option to publish the peer review history of their article (what does this mean?). If published, this will include your full peer review and any attached files.

Reviewer #1: No

Reviewer #2: No

---

## [Author Response · Author response to Decision Letter 0]

4 Aug 2023

Dear Dr. Muhammad Eid Akkawi

Academic Editor/ PLOS ONE

RE: Manuscript entitled "Potentially inappropriate medication uses and associated factors among elderly primary health care clinics attendees: a call to action" ([PONE-D-23-10547] - [EMID:c0429b4b8a8af4b0]). 

We appreciate your time and effort in evaluating the reviewers' comments and responding to us. 

We appreciate yours and the reviewers' comments. We carefully read each reviewer's comments and the 

Journal Requirements and responded to them point by point below.

We submitted one clean copy and a marked-up copy of our manuscript that highlights changes made to the original version. Our responses below are in the “Authors’ Response” column and shown in blue to distinguish them from the Reviewers’ comments. 

We hope that we have adequately responded to the reviewers' concerns. We believe that this has strengthened our paper and look forward to hearing from you. Please let us know if you have any additional suggestions or if anything is unclear.

Many thanks for considering our revision for publication.

Sincerely, 

Zaher Nazzal

---

## [Editor Report · Decision Letter 1]

8 Aug 2023

PONE-D-23-10547R1Potentially inappropriate medication uses and associated factors among elderly primary health care clinics attendees: a call to actionPLOS ONE

Dear Dr. Nazzal,

Thank you for submitting your manuscript to PLOS ONE. After careful consideration, we feel that it has merit but does not fully meet PLOS ONE’s publication criteria as it currently stands. Therefore, we invite you to submit a revised version of the manuscript that addresses the points raised during the review process.

We look forward to receiving your revised manuscript.

Kind regards,

Muhammad Eid Akkawi

Academic Editor

PLOS ONE

Journal Requirements:

Additional Editor Comments:

Abstract:

probably inappropriate medication (PIM), potentially not probably.

aged 65 and up >> aged 65 and above.

Sulfonylureas being the most common>> were the most common

Polypharmacy [aOR= 4.1, 95%CI: 1.6 – 10.7], the number in brackets are related to hyperpolypharmacy (> 10 drugs) while for polypharmacy it is [aP-value =.003, aOR= 2.8, 95%CI: 1.4–5.4]. Please choose which one you want it to appear in your abstract.

one-third of the elderly attending >> of the elderly patients attending

Improving PHC clinics' physicians' awareness >> Improving physicians' awareness

Introduction:

STOP to be corrected to STOPP throughout the manuscript.

“It was developed in 1991 by Mark Beers and has been updated every three years since 2011” not really every three years. Just change it to every few years.

benzodiazepine (14) >> benzodiazepines (14)

They are significantly affected by chronic diseases such as Diabetes, hypertension, congestive heart failure, atrial fibrillation, and coronary heart disease, most commonly with their medications furosemide, hydrochlorothiazide, and atorvastatin, levothyroxine, lisinopril, metoprolol, simvastatin, atenolol, amlodipine, and metformin>> Please paraphrase.

AGS>> should be written in full when it first appears.

Methodology:

clinics in the Ministry >> clinics of the Ministry

In addition, most of the Palestinian are elderly (two-thirds) had governmental insurance rather than private or no insurance >> not clear. Please paraphrase.

Results:

Most (90.5%) >> Most of them (90.5%)

The most common is hypertension >> The most common disease was hypertension

one-fourth have more than three >> one-fourth of them has more than three

Table 2: to change the caption to “Distribution of the most commonly prescribed medications…” as the table does not show all medications used under each drug class.

The prevalence of PMI >> PIM

Discussion:

One of the reviewer’s comments was not addressed, which is “Compare the results of the most encountered PIMs (SU, alpha blockers and NSAIDs) with other studies”. The only added paragraph was the comparison with another Palestinian study.

---

## [Author Response · Author response to Decision Letter 1]

11 Aug 2023

Dear Dr. Muhammad Eid Akkawi

Academic Editor/ PLOS ONE

RE: Manuscript entitled "Potentially inappropriate medication uses and associated factors among elderly primary health care clinics attendees: a call to action" ([PONE-D-23-10547] - [EMID:c0429b4b8a8af4b0]). 

We appreciate your time and effort in evaluating our revised manuscript and responding to us. 

We appreciate your comments and suggestions. We carefully read them and the Journal Requirements and responded to them point by point below.

We submitted one clean copy and a marked-up copy of our manuscript that highlights changes made to the original version. Our responses below are in the “Authors’ Response” column and shown in blue to distinguish them from the Reviewers’ comments. 

We hope that we have adequately responded to your comments. We believe that this has strengthened our paper and look forward to hearing from you. Please let us know if you have any additional suggestions or if anything is unclear.

Many thanks for considering our revision for publication.

Sincerely, 

Zaher Nazzal

---

## [Editor Report · Decision Letter 2]

14 Aug 2023

Potentially inappropriate medication uses and associated factors among elderly primary health care clinics attendees: a call to action

PONE-D-23-10547R2

Dear Dr. Nazzal,

We’re pleased to inform you that your manuscript has been judged scientifically suitable for publication and will be formally accepted for publication once it meets all outstanding technical requirements.

Kind regards,

Muhammad Eid Akkawi

Academic Editor

PLOS ONE

---

## [Editor Report · Acceptance letter]

16 Aug 2023

PONE-D-23-10547R2 

Potentially inappropriate medication uses and associated factors among elderly primary health care clinics attendees: a call to action 

Dear Dr. Nazzal:

I'm pleased to inform you that your manuscript has been deemed suitable for publication in PLOS ONE. Congratulations! Your manuscript is now with our production department. 

Kind regards, 

on behalf of

Dr. Muhammad Eid Akkawi 

Academic Editor

PLOS ONE